# Causal Context Adjustment Loss
# for Learned Image Compression

**Minghao Han[1], Shiyin Jiang[1], Shengxi Li[2], Xin Deng[2], Mai Xu[2], Ce Zhu[1], Shuhang Gu[1]***

[1]University of Electronic Science and Technology of China  [2]Beihang University

`{minghao.hmh, shuhanggu}@gmail.com`

## Abstract

In recent years, learned image compression (LIC) technologies have surpassed conventional methods notably in terms of rate-distortion (RD) performance. Most present learned techniques are VAE-based with an autoregressive entropy model, which obviously promotes the RD performance by utilizing the decoded causal context. However, extant methods are highly dependent on the fixed hand-crafted causal context. The question of how to guide the auto-encoder to generate a more effective causal context benefit for the autoregressive entropy models is worth exploring. In this paper, we make the first attempt in investigating the way to explicitly adjust the causal context with our proposed Causal Context Adjustment loss (CCA-loss). By imposing the CCA-loss, we enable the neural network to spontaneously adjust important information into the early stage of the autoregressive entropy model. Furthermore, as transformer technology develops remarkably, variants of which have been adopted by many state-of-the-art (SOTA) LIC techniques. The existing computing devices have not adapted the calculation of the attention mechanism well, which leads to a burden on computation quantity and inference latency. To overcome it, we establish a convolutional neural network (CNN) image compression model and adopt the unevenly channel-wise grouped strategy for high efficiency. Ultimately, the proposed CNN-based LIC network trained with our Causal Context Adjustment loss attains a great trade-off between inference latency and rate-distortion performance. The code is available here.

## 1 Introduction

The burgeoning quality of high-resolution photos is driving an increasing demand for advanced image storage and transmission technologies. Consequently, lossy image compression techniques have been growing extraordinarily fast in recent years. In parallel to conventional coding technologies such as JPEG [42], BPG [6], WebP [15], VVC [40], learned image compression (LIC) methods [3, 4, 10, 11, 17, 18, 20, 30, 35, 36, 47] emerge, achieving high peak signal-to-noise ratio (PSNR) and multiscale structural similarity (MS-SSIM) [44] while operating fairly fast. Their superior compression results over those of VVC demonstrate an enormous possibility that LIC technology would appear on par with the traditional ones in the near future.

Learned lossy image compression methods are built upon a variational auto-encoder (VAE) framework proposed by Ballé et al. [4]. The VAE based LIC framework mainly comprises an auto-encoder and an entropy model. The auto-encoder conducts nonlinear transforms between the image space and the latent representation space; while, the entropy model minimizes the code length by estimating the probability distribution of latent representations. In comparison to the auto-encoder, which could borrow ideas from recent advances in network architecture design, the entropy model is a unique important component to LIC and has a vital influence on the final compression results.

---

*Corresponding Author

38th Conference on Neural Information Processing Systems (NeurIPS 2024).

In the literature on LIC, the entropy model generally refers to a parameterized distribution model. In their seminal work [3], Ballé et al. established the end-to-end rate-distortion minimization framework and showed that the smallest average code length of latent representation is given by the Shannon cross entropy between the actual marginal distribution and a learned entropy model. Since then, numerous entropy models have been investigated. One category of studies investigates advanced network architectures for accurately predicting the distribution of latent representations. Meanwhile, another line of research study a more fundamental perspective of the entropy model, i.e. conditional distribution modeling, to pursue a better rate-distortion trade-off. Taking side information (also termed as hyperprior) and decoded latent (also termed causal context) as conditions has become a prevailing strategy in state-of-the-art LIC models.

In this paper, we advance conditional distribution modeling in the entropy model with our proposed causal context adjustment loss (CCA-loss). Existing works generally train LIC networks with a combination of the rate loss and the distortion loss. The conditional predictability of the representation is indirectly optimized, and the performance of entropy model highly relies on the hand-crafted causal context model, e.g. channel-wise [36], checkerboard [18] and space-channel [17] context model. Our CCA-loss makes the first attempt on explicitly imposing loss to adjust the causal context, making the latter representation more accurately predicted by the previously decoded representations. To be more specific, considering a two stage autoregresive context model with hyperprior $z$, denote the latent representation to be decoded in the first and second stage as $y_1$ and $y_2$; in addition to minimizing the cross entropy loss for reducing bitstream, we introduce an auxiliary entropy model and a tailored context causal adjustment loss, which let $y_2$ can be accurately estimated by $y_1$ and $z$, while, at the same time, let $y_2$ can not be accurately estimated by merely $z$. In this vein, our CCA-loss explicitly guides the encoder to adjust important information into the early stage of the autoregressive entropy model, providing the LIC framework a more rational causal context sequence for entropy coding. As the codes in the early stages are enhanced with our CCA-loss, we further study the schedule of causal context transmission, and adopt an uneven channel dimension schedule for the pursuit of a better rate-distortion trade-off. The uneven channel schedule is also beneficial for reducing computational burden in the coding and decoding process, enabling our model to achieve state-of-the-art compression performance with less running time. Our contributions are summarized as follows:

- We introduce causal context adjustment loss to explicitly adjust the causal context information, forcing the network to encode important information early and therefore improving the autoregressive prediction accuracy of the entropy model.

- We adopt an uneven schedule of autoregressive causal context and a convolutional auto-encoder architecture, delivering an efficient compression network which is easy to be implemented on modern deep learning platforms.

- We evaluate our proposed compression network on various benchmark datasets, in which our method achieves better rate-distortion trade-offs towards the existing state-of-the-art methods, with more than 20% less compression latency.

## 2 Related Works

Recent LIC studies broadly follow the seminal work of Ballé et al. [3], which utilizes a VAE based framework for rate-distortion optimization. Generally, VAE-based LIC models comprise an auto-encoder and an entropy model. In this section, we review respective progresses in advanced auto-encoders and entropy models. Another considerable technique in LIC is the quantization method, however, as we did not dig into the details and simply followed the quantization method of [35], we omit the review of quantization methods in this section.

### 2.1 Auto-Encoder Architectures for Learned Image Compression

The auto-encoder plays the role of extracting a latent representation apt to be compressed in LIC framework. In their pioneering work, Ballé et al. [3] first proposed to use a generalized divisive normalization (GDN) [2] to transform the input image into latent space. The later works followed the same VAE framework for LIC but exploited the convolution neural network (CNN) architecture, which is easier to implement and train. Beyond the basic CNN auto-encoder, the introduction of more

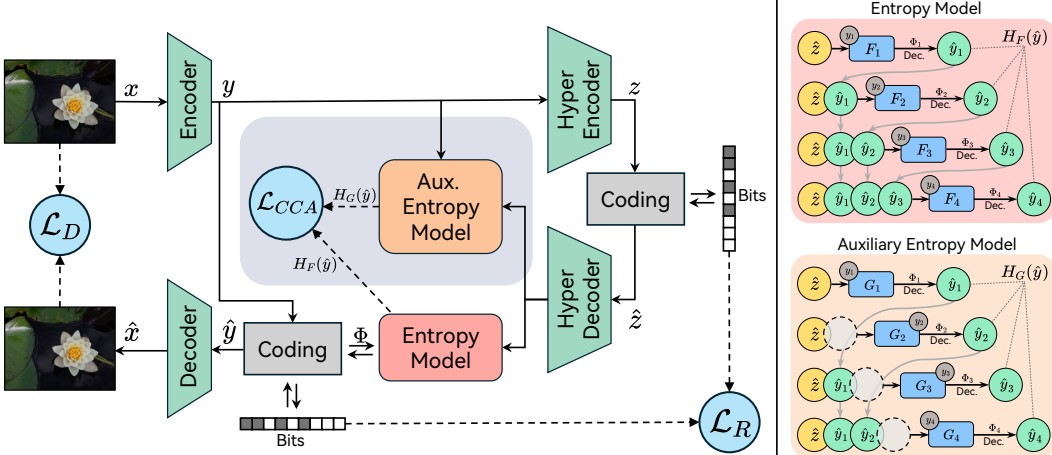

Figure 1: **Left**: A systematic overview of our method. We adopt the VAE-based framework [3] with hyperprior [4] and channel-wise autoregressive entropy model [35]; besides the original Rate-Distortion loss ($\mathcal{L}_R$, $\mathcal{L}_D$), we introduce an auxiliary entropy model and propose the causal context adjustment loss ($\mathcal{L}_{CCA}$) for better training the entropy model. **Right**: An illustration of the entropy model and the auxiliary entropy model. The auxiliary entropy model does not use the information to be encoded to predict the following representations, our $\mathcal{L}_{CCA}$ encourage the predicting gap between the two models, so as to enhance the importance of causal context in early stages.

complex nonlinear transforms [9, 11, 33] and various architectures [14, 28, 45, 46] promotes RD performance. Recently, inspired by the great successes Transformers have made in other vision tasks, self-attention modules have been widely utilized for extracting latent representations. Embedding Transformer variants, for example ViT [12], swin-Transformer [31] in the auto-encoder [32, 48, 49] enhances the RD performance. Moreover, Liu et al. [30] proposed a hybrid approach, combining conventional CNN and swin-Transformer. Although these Transformer-based auto-encoder could improve the RD performance by extracting better latent representations, the inference of transformer architecture has not been well optimized by the existing hardware, resulting in slow coding and decoding speed. In this paper, we borrow ideas from recent advances in image restoration [8] and adopt a CNN-based auto-encoder architecture. Thanks to our improved entropy model as well as the powerful NAF-block [8], our LIC model could achieve state-of-the-art compression results with much less runtime than recent Transformer-based approaches.

## 2.2 Entropy Models for Learned Image Compression

The entropy model plays a key role in LIC for minimizing the bitstream of latent representation. In the original work [3], the probability distributions of the latent representation are modeled using a non-parametric, fully factorized density model. In order to improve the distribution predicting accuracy, Ballé et al. [4] introduced side information as a hyperprior latent variable and ultimately established the basic VAE architecture of LIC in the past decade. Beyond the hyperprior transmitted in the VAE-based LIC framework, newly proposed extra side information transmitted from encoder to decoder promotes the compression performance as well [20, 43]. Moreover, Duan et al. [13] explored the hierarchical VAE structure with multiple hyperpriors.

In addition to the improvement on the side information, the introduction of causal context autoregression greatly promotes the RD performance of LIC, efficiently utilizing the information from decoded parts without a supererogatory amount of bits per pixel (bpp). Minnen et al. [35] proposed the first autoregressive structure, using the decoded spatial context to better estimate the current probability distribution. Numerous works attempt to establish an effective causal context for assistance in distribution estimations, such as channel-wise segmentation [36], checkerboard [18], unevenly grouping [17]. A very recent work [34] explored different strategies to selectively transmit tokens. However, a fixed hand-crafted causal context may not work well in diverse image distributions. In this work, we impose a loss to adjust the causal context in the training phase, allowing the network to achieve a more accurate probability estimation.

An efficient network structure of the entropy model remains critical for achieving high RD performance [26, 27, 29]. Apart from the basic causal context and hyperprior entered into the entropy model, more references benefit RD performance [11, 16, 17, 38]. Just as how Transformer performs in auto-encoder, the advantages of integrating the entropy model with Transformer are unearthed quickly. Previous works applied various Transformer blocks [20, 23, 25, 30, 37] to enhance features before entropy estimation. Following our modified architecture in auto-encoder, we embed the NAF-block in the entropy model to improve estimation accuracy.

## 3 Preliminary: Learned Image Compression with Variational Auto-Encoder

**Variational Auto-Encoder based Image Compression Framework.** Ever since the VAE architecture was established [21], learned lossy image compression techniques maintain the primary constituent structure [3], including an auto-encoder to extract the latent representation for compression and an entropy model to assist in entropy coding. Given a source image vector $\boldsymbol{x}$, the auto-encoder contains a parametric analysis transform $g_a$ to obtain the latent representation $\boldsymbol{y}$ from $\boldsymbol{x}$ and a parametric synthesis transform $g_s$ for reconstruction. $\boldsymbol{y}$ is then quantized to $\hat{\boldsymbol{y}}$, the discrete coding symbol for storage. The probability distributions of $\hat{\boldsymbol{y}}$ are modeled using a factorized density model $\boldsymbol{\psi}$ as $p_{\hat{\boldsymbol{y}}|\boldsymbol{\psi}}(\hat{\boldsymbol{y}}|\boldsymbol{\psi}) = \prod_i (p_{\boldsymbol{y}_i|\boldsymbol{\psi}}(\boldsymbol{\psi}) * \mathcal{U}(-\frac{1}{2}, \frac{1}{2}))(\hat{\boldsymbol{y}}_i)$. As quantization introduces error, which is tolerated in the context of lossy compression, the optimization target approximates the true posterior $p_{\hat{\boldsymbol{y}}|\boldsymbol{x}}(\hat{\boldsymbol{y}}|\boldsymbol{x})$ with a neural network $\tilde{q}(\hat{\boldsymbol{y}}|\boldsymbol{x})$ as the expectation of their Kullback-Leibler (KL) divergence over the data distribution $p_{\boldsymbol{x}}$:

$$\mathbb{E}_{\boldsymbol{x}\sim p_{\boldsymbol{x}}} D_{KL}[\tilde{q} \parallel p_{\hat{\boldsymbol{y}},\hat{\boldsymbol{z}}|\boldsymbol{x}}] = \mathbb{E}_{\boldsymbol{x}\sim p_{\boldsymbol{x}}}\mathbb{E}_{\hat{\boldsymbol{y}},\hat{\boldsymbol{z}}\sim\tilde{q}}\big[ -\log p_{\boldsymbol{x}|\hat{\boldsymbol{y}}}(\boldsymbol{x}|\hat{\boldsymbol{y}}) - \log p_{\hat{\boldsymbol{y}}|\boldsymbol{\psi}}(\hat{\boldsymbol{y}}|\boldsymbol{\psi})\big]. \tag{1}$$

The former term refers to the image reconstruction distortion (measured by PSNR or MS-SSIM), and the latter term represents the bit-rate (expected code length). A hyperparameter $\lambda$ is multiplied on the latter term, so that we can control the rate-distortion trade-off to obtain various compression rates.

**Entropy Model with Hyperprior.** However, directly modeling $\hat{\boldsymbol{y}}$ with the factorized density model $\boldsymbol{\psi}$ is less than satisfactory, as the estimation of which is not accurate and out of correlation with the data distributions. To capture the spatial dependence among the elements of $\hat{\boldsymbol{y}}$, the side information $\boldsymbol{z}$ is introduced [4]. $\boldsymbol{z}$ is generated by a hyper analysis transform $h_a$ from $\boldsymbol{y}$, transmitted as a hyperprior latent feature to help predict the distributions of $\hat{\boldsymbol{y}}$ accurately. Similarly to $\boldsymbol{y}$, $\boldsymbol{z}$ is quantized to $\hat{\boldsymbol{z}}$ in the same manner. The probability distributions of $\hat{\boldsymbol{z}}$ are calculated using a factorized density model $\boldsymbol{\psi}$, to encode $\hat{\boldsymbol{z}}$ as $p_{\hat{\boldsymbol{z}}|\boldsymbol{\psi}}(\hat{\boldsymbol{z}}|\boldsymbol{\psi}) = \prod_i (p_{\boldsymbol{z}_i|\boldsymbol{\psi}}(\boldsymbol{\psi}) * \mathcal{U}(-\frac{1}{2}, \frac{1}{2}))(\hat{\boldsymbol{z}}_i)$. During the entropy coding process, $\hat{\boldsymbol{z}}$ would be entered into a hyper synthesis transform $h_s$ to acquire the estimations $\{\boldsymbol{\mu}, \boldsymbol{\sigma}\}_i$ in normal distribution of each element $\hat{\boldsymbol{y}}_i$. This course can be formulated as $p_{\hat{\boldsymbol{y}}|\hat{\boldsymbol{z}}}(\hat{\boldsymbol{y}}|\hat{\boldsymbol{z}}) = \prod_i \big(\mathcal{N}(\boldsymbol{\mu}_i, \boldsymbol{\sigma}_i^2) * \mathcal{U}(-\frac{1}{2}, \frac{1}{2})\big)(\hat{\boldsymbol{y}}_i)$, with $\{\boldsymbol{\mu}, \boldsymbol{\sigma}\} = h_s(\hat{\boldsymbol{z}})$. The KL divergence in the basic VAE structure (Eq. 1) can be expanded as follows:

$$\mathbb{E}_{\boldsymbol{x}\sim p_{\boldsymbol{x}}} D_{KL}[\tilde{q} \parallel p_{\hat{\boldsymbol{y}},\hat{\boldsymbol{z}}|\boldsymbol{x}}] = \mathbb{E}_{\boldsymbol{x}\sim p_{\boldsymbol{x}}}\mathbb{E}_{\hat{\boldsymbol{y}},\hat{\boldsymbol{z}}\sim\tilde{q}}\big[ -\log p_{\boldsymbol{x}|\hat{\boldsymbol{y}}}(\boldsymbol{x}|\hat{\boldsymbol{y}}) - \log p_{\hat{\boldsymbol{y}}|\hat{\boldsymbol{z}}}(\hat{\boldsymbol{y}}|\hat{\boldsymbol{z}}) - \log p_{\hat{\boldsymbol{z}}|\boldsymbol{\psi}}(\hat{\boldsymbol{z}}|\boldsymbol{\psi})\big]. \tag{2}$$

On the other side of VAE, the parametric synthesis transform $g_s$ recovers the reconstructed image $\hat{\boldsymbol{x}}$ from the decoded $\hat{\boldsymbol{y}}$. Fig. 1 reveals the general basic structure.

**Autoregressive Entropy Model.** In addition, an advanced architecture of the entropy model is the joint autoregression [35], which soon develops into a more efficient channel-wise autoregression [36]. In the channel-wise autoregressive structure, the latent representation $\hat{\boldsymbol{y}}$ is grouped in the channel dimension and decoded in order. Thus, the second term of the KL divergence in hyperprior structure (Eq. 2) is expanded as:

$$\mathbb{E}_{\hat{\boldsymbol{y}},\hat{\boldsymbol{z}}\sim\tilde{q}}\big[ -\log p_{\hat{\boldsymbol{y}}|\hat{\boldsymbol{z}}}(\hat{\boldsymbol{y}}|\hat{\boldsymbol{z}})\big] = \mathbb{E}_{\hat{\boldsymbol{y}}_i,\hat{\boldsymbol{z}}\sim\tilde{q}}\big[ -\log p_{\hat{\boldsymbol{y}}_1|\hat{\boldsymbol{z}}}(\hat{\boldsymbol{y}}_1|\hat{\boldsymbol{z}})p_{\hat{\boldsymbol{y}}_2|\hat{\boldsymbol{z}},\hat{\boldsymbol{y}}_1}(\hat{\boldsymbol{y}}_2|\hat{\boldsymbol{z}},\hat{\boldsymbol{y}}_1)$$
$$p_{\hat{\boldsymbol{y}}_3|\hat{\boldsymbol{z}},\hat{\boldsymbol{y}}_1,\hat{\boldsymbol{y}}_2}(\hat{\boldsymbol{y}}_3|\hat{\boldsymbol{z}},\hat{\boldsymbol{y}}_1,\hat{\boldsymbol{y}}_2)\cdots p_{\hat{\boldsymbol{y}}_n|\hat{\boldsymbol{z}},\hat{\boldsymbol{y}}_1,\cdots,\hat{\boldsymbol{y}}_{n-1}}(\hat{\boldsymbol{y}}_n|\hat{\boldsymbol{z}},\hat{\boldsymbol{y}}_1,\cdots,\hat{\boldsymbol{y}}_{n-1})\big]. \tag{3}$$

Besides the prior of $\hat{\boldsymbol{z}}$, the estimation of the autoregressive entropy model conditions more on the causal context, that is, the model utilizes the information from the decoded parts (causal context) without introducing additional redundancy in information transmission. Therefore, the more effective the causal context, the stronger the performance of the autoregressive entropy model. Existing methods adopt various hand-crafted causal contexts to enhance it. We expect to establish a way that enables the network to adaptively adjust the causal context. Imposing a loss to explicitly adjust the causal context is a delicate way.

# 4 Causal Context Adjustment for Efficient Learned Image Compression

In this section, we introduce the details of our LIC method. We firstly introduce our causal context adjustment (CCA) loss, which is able to explicitly push the encoder to encode important (in terms of information gain) representation at an earlier stage for better predicting the remaining representations. Subsequently, we introduce the implementation details of our efficient LIC method, including our CNN-based encoder and decoder architecture, unevenly grouped autoregressive schedule, light-weight entropy model, and overall loss function.

## 4.1 Causal Context Adjustment

As introduced in the previous section, exploiting the causal context from the hyperprior and the autoregressive framework to establish a conditional distribution task is the key to a state-of-the-art entropy model. While introducing conditions is undoubtedly beneficial for improving the accuracy of distribution estimation, existing works intuitively set up the context models, such as checkerboard context model and slice-based context model, and there still lack in-depth study on how to constitute the causal context rationally in the literature. More concretely, the rate and distortion loss reflect the prediction error given the causal context and the reconstruction error given the decoded representations, respectively; neither of them could explicitly affect the organization of the causal context. In this section, we introduce the CCA-loss, which explicitly encourages important information of the image to be encoded into earlier causal context, so as to enhance the predictability of the autoregressive entropy model. To the best of our knowledge, our work is the first attempt that introduces loss instead of intuitively adjusting the context architecture to improve the context model.

To introduce our proposed Causal Context Adjustment (CCA) loss, we first revisit the hyperprior and the autoregressive entropy model. Without loss of generality, we consider a two-stage autoregressive context model. To encode the same latent representation, the cross entropy of the hyperprior model and the hyperprior + autoregressive model can be written as follows:

$$H_{\text{H.P.}}(q(\hat{\boldsymbol{y}}|\hat{\boldsymbol{z}}), p(\hat{\boldsymbol{y}}|\hat{\boldsymbol{z}})) = H(q(\hat{\boldsymbol{y}}_1|\hat{\boldsymbol{z}}), p(\hat{\boldsymbol{y}}_1|\hat{\boldsymbol{z}})) + H(q(\hat{\boldsymbol{y}}_2|\hat{\boldsymbol{z}}), p(\hat{\boldsymbol{y}}_2|\hat{\boldsymbol{z}})), \tag{4}$$

$$H_{\text{H.P.+A.R.}}(q(\hat{\boldsymbol{y}}|\hat{\boldsymbol{z}}), p(\hat{\boldsymbol{y}}|\hat{\boldsymbol{z}})) = H(q(\hat{\boldsymbol{y}}_1|\hat{\boldsymbol{z}}), p(\hat{\boldsymbol{y}}_1|\hat{\boldsymbol{z}})) + H(q(\hat{\boldsymbol{y}}_2|\hat{\boldsymbol{z}}, \hat{\boldsymbol{y}}_1), p(\hat{\boldsymbol{y}}_2|\hat{\boldsymbol{z}}, \hat{\boldsymbol{y}}_1)), \tag{5}$$

where $H_{\text{H.P.}}$ and $H_{\text{H.P.+A.R.}}$ represent the cross entropy with hyperprior and with hyperprior + autoregressive estimation, respectively; $q$ and $p$ denotes the real distribution and the learned entropy model. According to Shannon information theory [39], as more information is incorporated in the estimation of $\hat{\boldsymbol{y}}_2$, $H_{\text{H.P.+A.R.}}$ is less than or equal to $H_{\text{H.P.}}$. Moreover, the gap between $H_{\text{H.P.}}$ and $H_{\text{H.P.+A.R.}}$ is related to the amount of information $\hat{\boldsymbol{y}}_1$ could provide for estimating $\hat{\boldsymbol{y}}_2$. Therefore, by calculating the following equation:

$$H_{\text{H.P.}} - H_{\text{H.P.+A.R.}} = H(q(\hat{\boldsymbol{y}}_2|\hat{\boldsymbol{z}}), p(\hat{\boldsymbol{y}}_2|\hat{\boldsymbol{z}})) - H(q(\hat{\boldsymbol{y}}_2|\hat{\boldsymbol{z}}, \hat{\boldsymbol{y}}_1), p(\hat{\boldsymbol{y}}_2|\hat{\boldsymbol{z}}, \hat{\boldsymbol{y}}_1)), \tag{6}$$

we could obtain the information gain introduced by causal context $\hat{\boldsymbol{y}}_1$. The above analysis inspires us to explicitly optimize Eq. 6 to enhance $\hat{\boldsymbol{y}}_1$, encouraging it to encode important information that helps to better estimate $\hat{\boldsymbol{y}}_2$. Concretely, in addition to the original entropy model $F(\hat{\boldsymbol{y}}_1, \hat{\boldsymbol{z}}) \rightarrow \hat{\boldsymbol{y}}_2$, we introduce an auxiliary entropy model, which only takes $\hat{\boldsymbol{z}}$ as input: $G(\hat{\boldsymbol{z}}) \rightarrow \hat{\boldsymbol{y}}_2$. With the introduced auxiliary entropy model, our CCA-loss can be defined as follows:

$$I(\hat{\boldsymbol{y}}_2; \hat{\boldsymbol{y}}_1) = \mathbb{E}_{\hat{\boldsymbol{y}}_1 \sim p_{\hat{\boldsymbol{y}}_1|\hat{\boldsymbol{z}}}} \mathbb{E}_{\hat{\boldsymbol{z}} \sim p_{\hat{\boldsymbol{z}}|\psi}} \big[ -\log p_{\hat{\boldsymbol{y}}_2|\hat{\boldsymbol{z}}}(\hat{\boldsymbol{y}}_2|\hat{\boldsymbol{z}}) + \log p_{\hat{\boldsymbol{y}}_2|\hat{\boldsymbol{z}}, \hat{\boldsymbol{y}}_1}(\hat{\boldsymbol{y}}_2|\hat{\boldsymbol{z}}, \hat{\boldsymbol{y}}_1) \big], \tag{7}$$

where $p_{\hat{\boldsymbol{y}}_2|\hat{\boldsymbol{z}}}(\hat{\boldsymbol{y}}_2|\hat{\boldsymbol{z}})$ and $p_{\hat{\boldsymbol{y}}_2|\hat{\boldsymbol{z}}, \hat{\boldsymbol{y}}_1}(\hat{\boldsymbol{y}}_2|\hat{\boldsymbol{z}}, \hat{\boldsymbol{y}}_1)$ are the estimated distributions of auxiliary and the major entropy model, respectively. The auxiliary and major entropy models are parameterized by two networks, i.e. $G(\hat{\boldsymbol{z}})$ and $F(\hat{\boldsymbol{y}}_1, \hat{\boldsymbol{z}})$. It should be noted that the auxiliary entropy model is only introduced in the training phase for better optimizing the causal context; in the testing phase, our model still uses $F(\hat{\boldsymbol{y}}_1, \hat{\boldsymbol{z}})$ to compress the latent representation, our CCA-loss will not introduce additional computational burden for image compression.

An illustration of the proposed CCA-loss can be found in Fig. 1. Besides the above analysis in Eq. 4 to Eq. 7, a straightforward interpretation of our CCA-loss is enlarging the prediction gap between the major entropy model $F(\hat{\boldsymbol{y}}_1, \hat{\boldsymbol{z}})$ and the auxiliary entropy model $G(\hat{\boldsymbol{z}})$; so that the encoder is forced to adjust causal context $\hat{\boldsymbol{y}}_1$ and make it contain important information for conditional modeling. For autoregressive models with more than two stages, Eq. 7 can be extended easily. Denote the $i$-th stage entropy model as $p_{\hat{\boldsymbol{y}}_i|\hat{\boldsymbol{z}}, \hat{\boldsymbol{y}}_{<i}}(\hat{\boldsymbol{y}}_i|\hat{\boldsymbol{z}}, \hat{\boldsymbol{y}}_{<i})$, which is parameterized with network

$F_i(\hat{z}, \hat{y}_{<i})$; we introduce the corresponding auxiliary entropy model $p_{\hat{y}_i|\hat{z}, \hat{y}_{<i-1}}(\hat{y}_i|\hat{z}, \hat{y}_{<i-1})$, which is parameterized with network $G_i(\hat{z}, \hat{y}_{<i-1})$. The multi-stage CCA-loss can be defined as follows:

$$\mathcal{L}_{CCA} = \sum_i \mathbb{E}_{\hat{y} \sim p_{\hat{y}|z}} \mathbb{E}_{\hat{z} \sim p_{\hat{z}|\psi}} \big[ -\log p_{\hat{y}_i|\hat{z}, \hat{y}_{<i-1}}(\hat{y}_i|\hat{z}, \hat{y}_{<i-1}) + \log p_{\hat{y}_i|\hat{z}, \hat{y}_{<i}}(\hat{y}_i|\hat{z}, \hat{y}_{<i}) \big]. \quad (8)$$

With our proposed CCA-loss, the learned image compression model is able to spontaneously adjust the causal context, thereby promoting the rate-distortion performance.

## 4.2 Training Compression Network with CCA Loss

### 4.2.1 Auto-Encoder Architecture

Inspired by the recent work [8], which designed a CNN-based nonlinear activation-free network to improve image restoration performance, we stack NAF-Blocks [8] in the analysis transform $g_a$ and the synthesis transform $g_s$. Following the previous CNN-based model [9, 17], we adopt the stacking residual blocks [19] in the auto-encoder transform for better nonlinearity. Due to the simplicity of the information that hyperprior $z$ carries, there are only simple convolution layers for the hyper analyzer $h_a$ and synthesizer $h_s$. Thanks to our convolutional architecture, our approach is much faster than recent LIC methods which generally adopt Transformer blocks to comprise the auto-encoder. Detailed architectures of our auto-encoder can be found in the Supplementary Materials.

### 4.2.2 Channel-wise Unevenly Grouped Entropy Model

To establish a robust causal context and efficiently exploiting it in the autoregressive entropy models is the key to reaching state-of-the-art. The existing approaches generally constitute the causal context model intuitively. In this paper, we propose the causal context adjustment loss (CCA-loss), which compels the analysis transform to generate a more potent causal context, that is, the enhanced estimation gain of early-stage context towards the latter latent representation. Theoretically, our proposed CCA-loss is architecture-agnostic and can be utilized to train various encoders to adjust the causal context according to the given conditional modeling architecture. However, compared to the checkerboard context model that leverages adjacent spatial information as context, it is easier for our convolutional encoder to adjust information across feature channels. We therefore adopt a channel-wise grouped autoregressive architecture to design our entropy model. Furthermore, since our CCA-loss could explicitly adjust the significant information into the earlier channels, we explore an unevenly grouped strategy to take full advantage of the first several informative channels. On account of the accumulated contexts $[\hat{y}_1, \hat{y}_2, \cdots, \hat{y}_{i-1}]$ as input to the autoregressive entropy model to predict $\hat{y}_i$, the unevenly grouped strategy also brings us advantages in the number of parameters and run time. Following our auto-encoder structure, we also utilize NAF-blocks [8] for a superior trade-off between accuracy and speed. For detailed network architectures of our entropy model as well as auxiliary entropy model, please refer to our Supplementary Materials. The comprehensive analysis of the benefits of the evenly and unevenly grouped strategies brought by our CCA-loss will be presented in the ablation study section.

### 4.2.3 Overall Loss Function

We follow the commonly used rate-distortion optimization framework to train our model. In addition to the rate losses $\mathcal{R}(\hat{y})$, $\mathcal{R}(\hat{z})$ and the distortion loss $\mathcal{D}(\hat{x}, x)$, our proposed CCA-loss is introduced to explicitly adjust the causal context. The implementation of our CCA requires a group of auxiliary entropy models. In order to obtain feasible auxiliary entropy models, we further introduce auxiliary losses $\mathcal{L}_{Aux}$, which let the auxiliary model to estimate the same latent representation $\hat{y}$ as the major entropy model. Therefore, the overall losses used for training our models are listed as follows:

$$\mathcal{L} = \lambda \cdot [\mathcal{R}(\hat{y}) + \mathcal{R}(\hat{z})] + \mathcal{D}(\hat{x}, x) + \mathcal{L}_{CCA} + \mathcal{L}_{Aux}, \quad (9)$$

we only use one parameter $\lambda$ to adjust the compression rate. Detailed ablation studies about the introduced CCA-loss will be presented in our experimental section.

# 5 Experiments

## 5.1 Experimental Settings

**Datasets.** We follow the previous work [49] and train our models on the Open Images [24] dataset. Open Images Dataset contains 300k images with short edge no less than 256 pixels. For evaluation, three benchmarks, i.e., Kodak image set [22], Tecnick test set [1] , and CLIC professional validation dataset [41], are utilized to evaluate the proposed network.

**Implementation details.** We set the channel of latent representation $\boldsymbol{y}$ as 320 and that of hyperprior $\boldsymbol{z}$ is set as 192. Following the previous works, we turn the quantization operation to $\lceil \boldsymbol{y} - \boldsymbol{\mu} \rfloor$ instead of $\lceil \boldsymbol{y} \rfloor$ and restore $\hat{\boldsymbol{y}}$ as $\lceil \boldsymbol{y} - \boldsymbol{\mu} \rfloor + \boldsymbol{\mu}$, which benefits the entropy models. We adopt the unevenly grouped strategy to segment the latent representation into 5 uneven slices. Our detailed unevenly grouped method and discussion on it can be found in the Supplementary Materials. Our experiments and evaluations are carried out on Intel Xeon Platinum 8375C and a single Nvidia RTX 4090 graphics card. We train our network with Adam optimizer. We randomly crop $256 \times 256$ sub-blocks from the Open Images dataset [24] with a batch size of 8. We optimize the network with the initial learning rate $1e - 4$ for 2M steps and then decrease the learning rate to $1e - 5$ for another 0.4M steps. The network is optimized with the MSE metric, which represents the distortion loss $\mathcal{D}$ in Eq. 9. For the MSE metric, the multipliers $\lambda$ before rate loss are $\{0.3, 0.85, 1.8, 3.5, 7, 15\}$.

**Comparison methods and metrics.** We compare our method with the hand-crafted coding standards VVC [40], BPG [6] and WebP [15] and recent state-of-the-art methods [4, 11, 17, 30, 45, 49]. The results of hand-crafted methods and Ballé2018 [4] are based on the implementation from CompressAI [5], while, the results of other methods are provided by the method authors. We mainly use PSNR to evaluate the image quality of compression results and use bits per pixel (bpp) value to indicate the compression ratio. The BD-rate [7] and runtime of several methods are also reported to comprehensively evaluate our model. Following the commonly used setting, we also compare the MS-SSIM metric on the Kodak dataset, the MS-SSIM optimized results by different methods are shown in our Supplementary Materials.

## 5.2 Ablation Study

We firstly conduct ablation experiments to validate the effectiveness of the proposed CCA-loss. In order to facilitate the analysis, we establish a tiny model to conduct our ablation experiments. We halve the channel number and stacking count of NAF-blocks [8] in our model and only adopt a three-stage autoregressive entropy model. We evaluate our CCA-loss on evenly grouped channel-wise autoregressive model as well as unevenly grouped channel-wise autoregressive model. The BD-rates of different models are reported in Table 1, without any additional computation in the testing phase, our CCA-loss could improve the evenly grouped and unevenly grouped models by a considerable margin. Especially for the case of unevenly grouped strategy, which adopts a more aggressive strategy and decodes less number of channels in the early stage, the enhancement brought by our CCA-loss is quite large. The phenomena reveals that utilizing small amount of significant information as the initial condition is beneficial for autoregressive entropy modeling, which is in line with the motivation of our paper.

To further investigate the impact on information distributions of our proposed CCA-loss, we extend a visualization of the quantities of information (code length) in the hyperprior and latent representation. The averaged information distributed ratios on the Kodak testing images by different models are shown in Fig. 2. As can be clearly found in the histogram, for entropy models with the same network architecture, our CCA-loss is able push the network to encode significant information at an earlier stage of the autoregressive model.

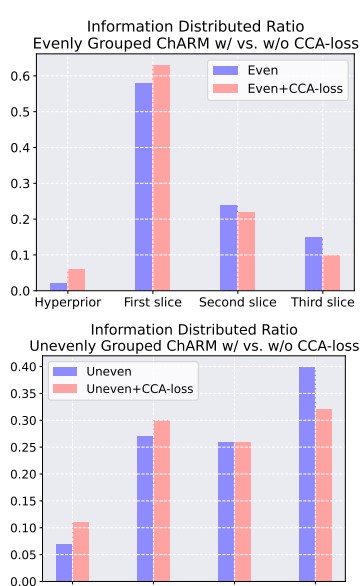

Figure 2: The comparison of averaged information distributed ratios of various models in Table 1.

Table 1: Experiments on Kodak dataset. The effects of our proposed Causal Context Adjustment loss (CCA-loss) are verified on various channel-wise autoregressive models. Note that the anchor BD-rate is set as the results of BPG evaluated on Kodak dataset (BD-rate = 0%).

| Model | CCA Loss (proposed) | Inference Time(ms) | BD-rate |
|---|---|---|---|
| ChARM (even) | | 126 | -13.31% |
| ChARM (even) | ✓ | 126 | -14.72% |
| ChARM (uneven) | | 116 | -14.56% |
| ChARM (uneven) | ✓ | 116 | -17.17% |
| BPG | - | - | 0% |

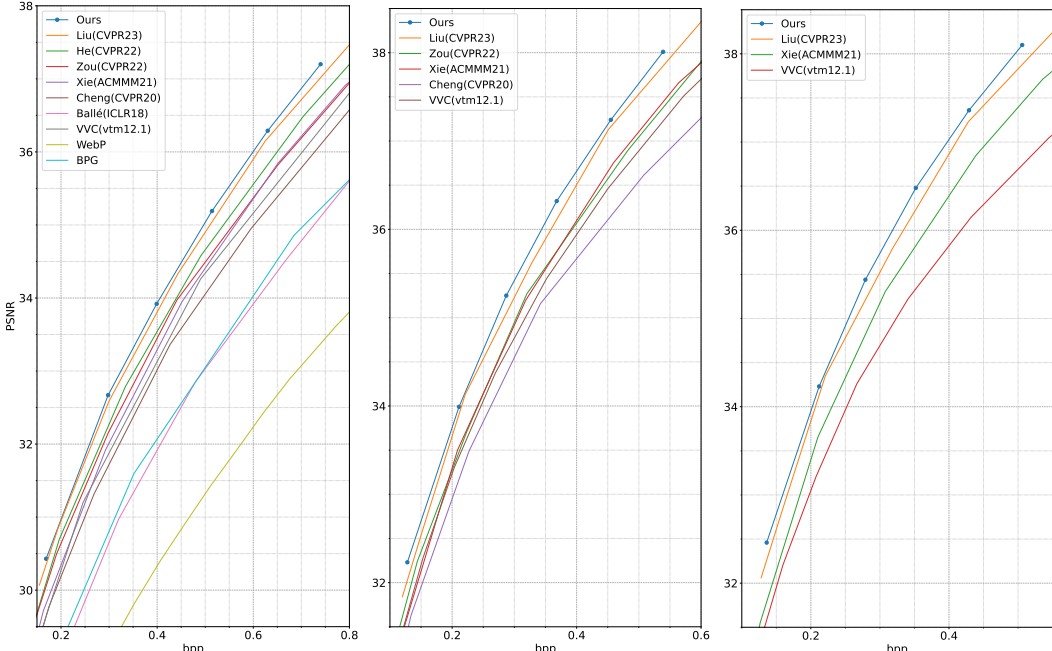

Figure 3: Rate-Distortion performance evaluation of PSNR on Kodak dataset (left), CLIC Professional Validation dataset (middle), Tecnick dataset (right), respectively.

## 5.3 Comparison with State-of-the-art Methods

**Rate-Distortion Comparison.** We evaluate the rate-distortion performance of our proposed models by drawing the rate-distortion curves. The distortion is assessed by PSNR while the rate is calculated by the bits per pixel (bpp). We first compare our proposed network with hand-crafted codec methods [6, 15, 40] and the LIC models that once reached state-of-the-art (SOTA) [4, 11, 17, 30, 45, 49] on the Kodak dataset. The result of the PSNR metric is presented in Fig. 3 (left), which demonstrates that our proposed methods could outperform other SOTA methods. The middle sub-figure and the right sub-figure in Fig. 3 are evaluated on the CLIC Professional Validation dataset and the Tecnick dataset, respectively. The SOTA results in various datasets show the generalization and robustness of our proposed model.

**Compression Latency.** As described in our introduction, we established a convolutional compression model for the pursuit of efficient compression. In Table 2, we present the coding latency, as well as the number of parameters and GFLOPs, by our proposed network and recent state-of-the-art methods [30, 48, 49]. The BD-rate values by different methods are also provided for reference, the anchor RD performance of which is set as the results of VVC (vtm-12.1) on Kodak dataset (BD-rate = 0%). As can be found in the table, our method achieves a better trade-off between compression performance and coding latency than the competing methods. With more than 20% less runtime, our model obtains about 2% BD-rate gain over [30].

Table 2: Comparison of coding complexity evaluated on Kodak dataset. All the models are evaluated on the same platform. The lower BD-rate is better.

| Model | Inference Latency(ms) | | | #Params | FLOPs(G) | BD-Rate |
|---|---|---|---|---|---|---|
| | Tot. | Enc. | Dec. | | | |
| Zou et al. [49] | 424 | 248 | 176 | 99.83M | 200.11 | -4.01% |
| Zhu et al. [48] | 272 | 129 | 143 | 56.93M | 364.08 | -3.00% |
| Liu et al. [30] | 255 | 122 | 133 | 75.90M | 700.65 | -11.88% |
| Ours | 201 | 109 | 92 | 64.89M | 615.93 | -13.87% |
| VVC | - | - | - | - | - | 0% |

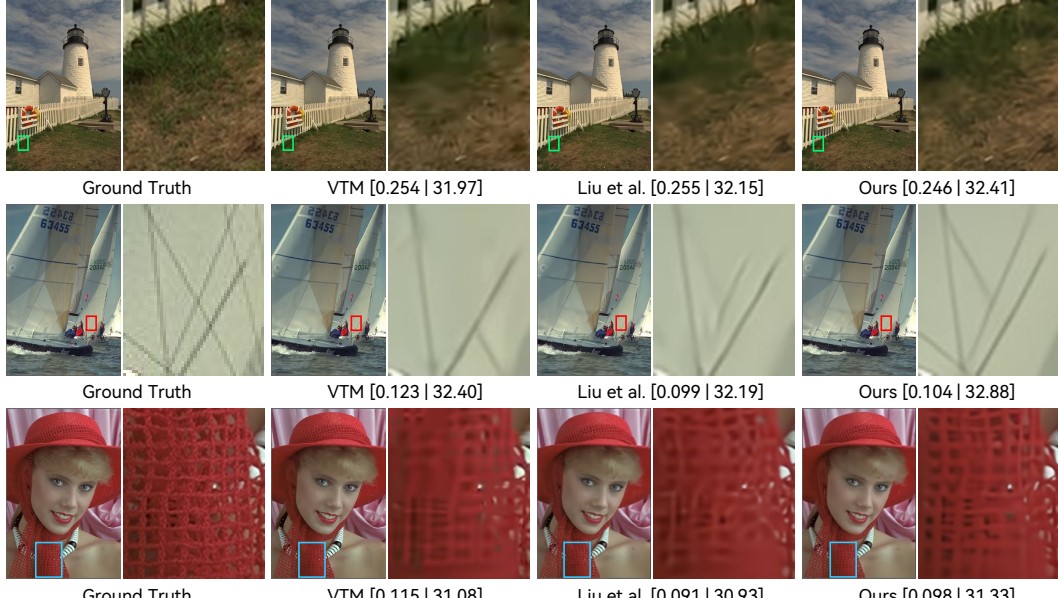

Figure 4: Visualization of the reconstructed images (top: *kodim19*, middle: *kodim10*, bottom: *kodim4*) from Kodak dataset. The titles under the sub-figures are represented as [bpp | PSNR(dB)].

**Visualization Analysis.** Our proposed learned image compression technology is capable of restoring the image details. Fig. 4 shows two sets of comparisons with the reconstruction of VVC [40] and a recent SOTA model [30]. The visualization results are produced at low bit-rates on the Kodak dataset [22]. The comparison of the reconstructed images demonstrates that our model restores more detailed and complicated textures than other methods. For example, we restore more sharp textures on the hat (*kodim4*), more details of the grassland (*kodim19*) and wrinkles on the sails (*kodim10*).

## 6 Conclusion

In this work, we explore the approach to adjust the causal context, which enables a superior channel-wise autoregressive model and more accurate estimation in probability distributions. By imposing the Causal Context Adjustment loss (CCA-loss) and the unevenly channel-wise grouped strategy on our proposed CNN-based model, we achieve state-of-the-art rate-distortion performance. Thanks to the advantages of convolutional neural network, our discussed unevenly grouped schedule and the training method by the proposed CCA-loss, our learned image compression model maintains a great trade-off between compression latency and RD performance. Furthermore, since we did not dive into the information redistributed phenomenon brought by the unevenly grouped strategy and CCA-loss training in this paper, the issue of the laws about the information distributed among the latent representation to be compressed is still worth investigating in the future.

## Acknowledgement

This work was supported by National Natural Science Foundation of China (No. 62250001, 62231002, 62020106011), Beijing Natural Science Foundation (No. L223021) and Sichuan Natural Science Foundation (No. 2024NSFTD0041).

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

# A  Network Architecture

## A.1  Architecture of transform networks

Table 3: Architecture of main transforms and hyper transforms.

| Analyzer $g_a$ | Synthesizer $g_s$ | Hyper Analyzer $h_a$ | Hyper Synthesizer $h_s$ |
|:---:|:---:|:---:|:---:|
| Conv 5×5, $\dim_0$, s2 | TConv 5×5, $\dim_2$, s2 | Conv 5×5, $\dim_2$, s2 | TConv 5×5, $\dim_2$, s2 |
| ResidualBlock×3 | NAF-Block×4 | GELU | GELU |
| NAF-Block×4 | ResidualBlock×3 | Conv 5×5, $\dim_2$, s2 | TConv 5×5, $\dim_2$, s2 |
| Conv 5×5, $\dim_1$, s2 | TConv 5×5, $\dim_1$, s2 | GELU | GELU |
| ResidualBlock×3 | NAF-Block×4 | Conv 5×5, 192, s2 | TConv 5×5, 320, s2 |
| NAF-Block×4 | ResidualBlock×3 | | |
| Conv 5×5, $\dim_2$, s2 | TConv 5×5, $\dim_0$, s2 | | |
| ResidualBlock×3 | NAF-Block×4 | | |
| NAF-Block×4 | ResidualBlock×3 | | |
| Conv 5×5, M, s2 | TConv 5×5, 3, s2 | | |

As introduced in our main paper, our compression framework is adopt the VAE framework proposed by Ballé et al. [3], and use the same strategy of hyperprior [4] and autoregressive entropy model [35]. Generally, the transform network comprise an analyzer $g_a$ and a synthesizer $g_s$, which play the role of feature extraction and image reconstruction. For extracting side information, another pair of hyper analyzer $h_a$ and hyper synthesizer $h_s$ is used to extracting and reconstructing the hyperprior variable $z$. The detailed network architecture of the above components can be found in Table 3.

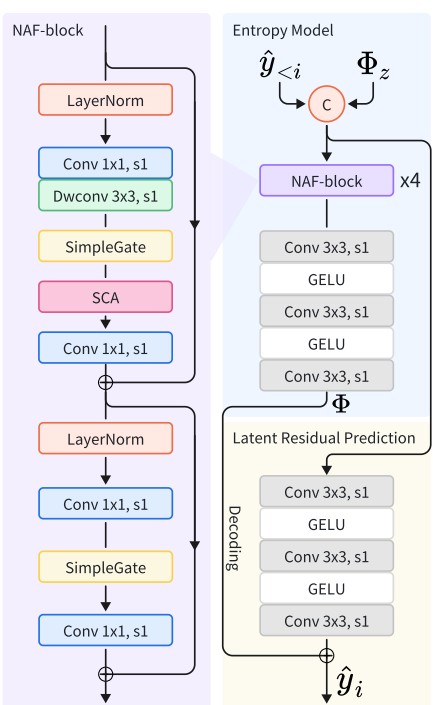

The 5×5 convolution and the 5×5 transposed convolution are utilized to downsample and upsample the feature maps, respectively. Following the previous works [9, 17], we adopt the commonly used residual blocks and the newly proposed NAF-Blocks to establish the analyzer and synthesizer. While, due to the simplicity of the information that hyperprior $z$ carries, there are only simple convolution layers for the hyper analyzer $h_a$ and hyper synthesizer $h_s$. The dimension numbers $\dim_0$, $\dim_1$ and $\dim_2$ in the table are set as 192, 224 and 256, respectively.

## A.2  Architecture of entropy model

Our proposed entropy model utilizes the NAF-block as well (see Fig. 5). The stacking NAF-blocks can enhance the concatenated features input to the entropy model, in order to obtain a more accurate estimation of the latent representation. The dimension of the latent representation in NAF-block is set as 224. Please note that we do not conduct the special training strategy like [8] for the simple channel attention (SCA) in the NAF-block, on account of no performance loss caused by this. Following the previous work [36], we append the latent residual prediction (LRP) to restore the error introduced by the quantization operation. For our auxiliary entropy model, the only difference is that the input removes the previous one slice, that is, replace $\hat{y}_{<i}$ with $\hat{y}_{<i-1}$ in Fig. 5.

Figure 5: Architecture of NAF-block, Entropy Model and Latent Residual Prediction (LRP).

# B  Adjustable Unevenly Grouped Strategy

To take full advantage of our proposed CCA-loss, we adopt the unevenly grouped strategy proposed by He et al. [17] in our method. In this part, We dive deeper into the specific grouped method and the advantages it brings. For the channel-wise autoregressive entropy models, the input to them is accumulated as the decoding progresses to the latter slices.

**Efficiency.**  For the $i$-th stage entropy model, the causal context contains $[\hat{\boldsymbol{y}}_1, \hat{\boldsymbol{y}}_2, \hat{\boldsymbol{y}}_3, \cdots, \hat{\boldsymbol{y}}_{i-1}] \in \mathbb{R}^{H \times W \times \sum_1^{i-1} C_i}$, where $H \times W$ is the spatial size of the latent representation and $C_i$ denotes the channel number of $\hat{\boldsymbol{y}}_i$. For the overall $n$-stage entropy model, the shape of the total causal context is written as $\sum_2^n \mathbb{R}^{H \times W \times \sum_1^{i-1} C_i}$, which can be expanded as $\sum_1^{n-1} \mathbb{R}^{H \times W \times (n-i)C_i}$. From this expression, we could see that the former slices are reused more times, leading to more parameters and latency. Thus, the unevenly grouped strategy could benefit the model in complexity, as Table 1 shows in the ablation study section.

**Rate-Distortion Trade-off.**  In this part, we analyze the effects of selecting different grouped schedules. Inspired by previous work [34], we parameterize the unevenly grouped strategy via a power schedule, i.e., $C(i) = N_{k,n} \cdot i^k$, where $k$ denotes the steepness of the increasing slices and $N$ normalizes the $n$-stage autoregressive slices in sum of $M$. For selecting best schedule for our model, we train compression models with different grouping hyperparameters with the loss function in Eq. 9 (including our CCA-loss). We evaluate grouping strategies with different $k$ values. The rate-distortion trade-offs by different models are shown in Fig. 6, the RD curve achieved by our selected schedule (i.e. $k = 1.7$) is presented for reference. As can be seen in the figure, the setting of $k = 1.7$ achieves the best rate-distortion performance, which we select for our ultimate SOTA model.

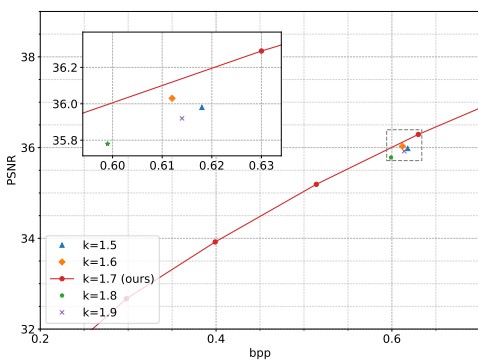

Figure 6: Compression results with different grouped schedules. Detailed experimental settings can be found in the main text.

# C  MS-SSIM Optimized Result

For higher MS-SSIM performance to adapt the real eyesight, we also produce the model of the MS-SSIM optimized objective. The distortion loss is replaced by $1 - \text{MS-SSIM}(\hat{\boldsymbol{x}})$ and the multiplier $\lambda$ before the rate loss are set as $\{0.2, 0.65, 1.5, 3.2, 6, 15\}$. The comparison of the rate-distortion curves with previous works is released in Fig. 7.

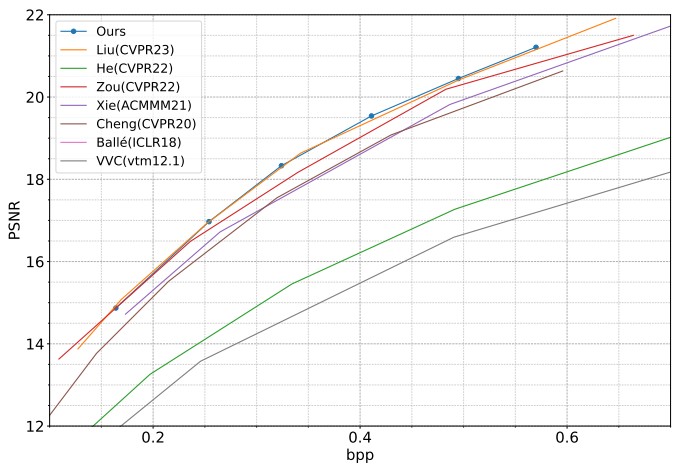

Figure 7: Rate-Distortion performance evaluation of MS-SSIM on Kodak dataset.

# D Image Reconstruction Visualization

We compare the reconstruction results on *kodim20* (Fig. 8) and *kodim24* (Fig. 9) of our model with those of Liu et al. [30] and several hand-crafted methods, i.e., VVC [40], Webp [15], JPEG [42].

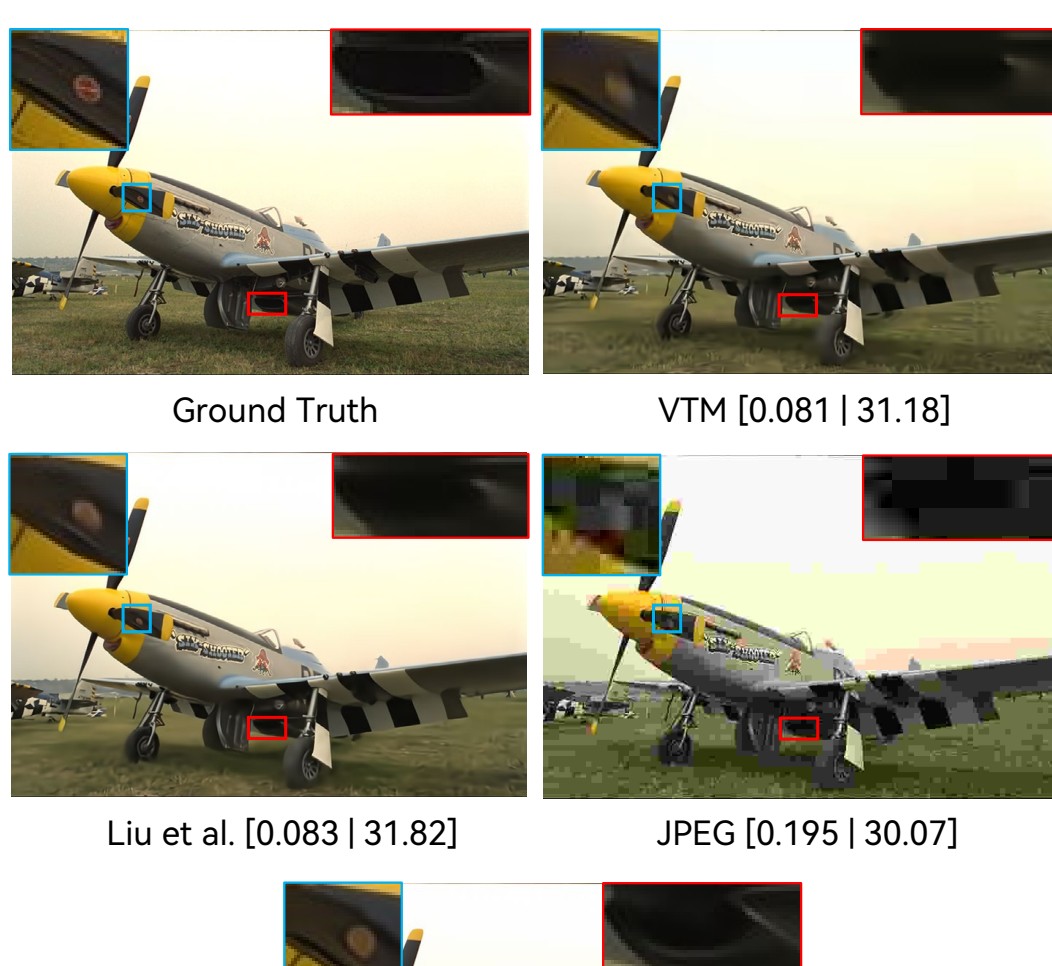

Ground Truth
VTM [0.081 | 31.18]

Liu et al. [0.083 | 31.82]
JPEG [0.195 | 30.07]

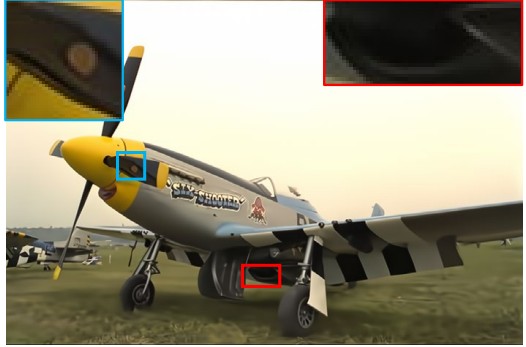

Ours [0.083 | 32.07]

Figure 8: Visual comparison on reconstructed propeller airplane (*kodim20*) image.

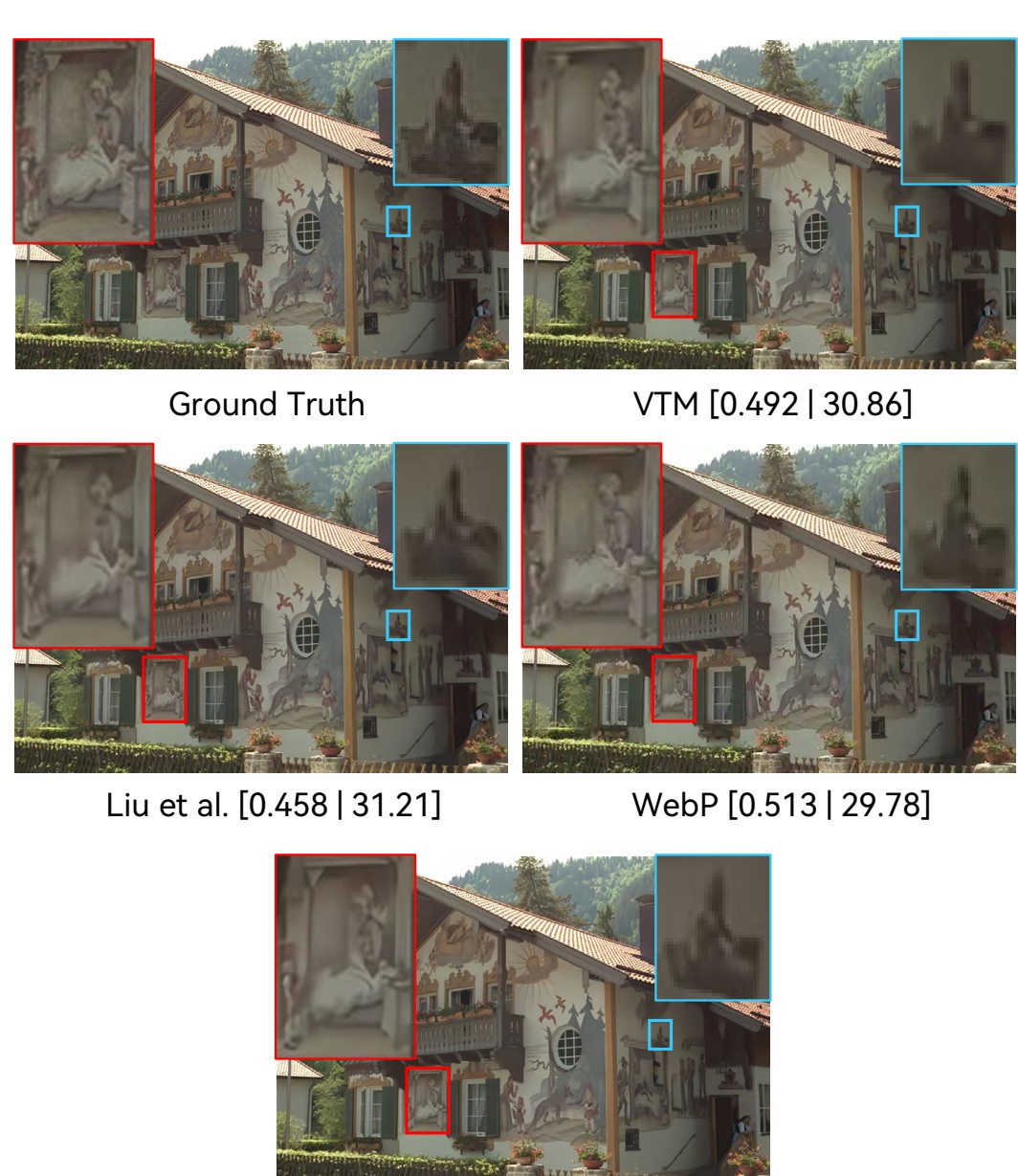

Figure 9: Visual comparison on reconstructed pattern on the walls of the house (*kodim24*) image.

