# OpenReview forum: "Causal Context Adjustment Loss for Learned Image Compression"
_NeurIPS.cc/2024/Conference — NeurIPS 2024 poster_

### Official Review · Reviewer_YBcc · 2024-07-06

**Soundness:** 3
**Presentation:** 3
**Contribution:** 3
**Rating:** 8
**Confidence:** 5

**Summary:**

The paper presents a novel approach to learned image compression (LIC) by introducing a Causal Context Adjustment loss (CCA-loss). This method aims to improve the rate-distortion (RD) performance of autoregressive entropy models used in LIC. The proposed approach allows the neural network to adjust the causal context dynamically, enhancing the accuracy of latent representation estimations. The paper also leverages a convolutional neural network (CNN) based model and an unevenly channel-wise grouped strategy to achieve a balance between inference latency and RD performance. Experimental results on benchmark datasets demonstrate that the proposed method outperforms existing state-of-the-art LIC techniques in both RD performance and computational efficiency.

**Strengths:**

1) Innovative Loss Function: The introduction of the CCA-loss represents a novel contribution that explicitly adjusts the causal context, leading to significant improvements in the accuracy of autoregressive entropy models.
2) Efficient Architecture: The proposed CNN-based model with uneven channel-wise grouping demonstrates significant improvements in both rate-distortion performance and computational efficiency.
3) Comprehensive Evaluation: The paper provides thorough experimental validation on multiple benchmark datasets, demonstrating superior RD performance and efficiency over state-of-the-art methods.
4) Clear Contributions: The paper clearly articulates its contributions, including the development of CCA-loss, the efficient CNN-based architecture, and the evaluation of these innovations through rigorous experiments.
5) Practical Implications: The reduction in compression latency by over 20% compared to state-of-the-art methods highlights the practical applicability of the proposed method.

**Weaknesses:**

1) Limited Exploration of Context Models: The paper mentions the potential for further investigation into the organization of causal contexts, indicating that current methods are somewhat intuitive and may benefit from more structured approaches.
2) Complexity in Implementation: While the proposed method is efficient, implementing the uneven channel-wise grouped strategy and the CCA-loss function may pose challenges for practitioners not well-versed in deep learning techniques.

**Questions:**

1) Further Exploration of Causal Context Models:
a) Question: The paper mentions that current methods for organizing causal contexts are somewhat intuitive. Can you elaborate on the potential for more structured approaches in organizing causal contexts? Are there any preliminary ideas or experiments that could guide future research in this direction?
b) Suggestion: Consider conducting a more detailed analysis or ablation study on different causal context models. This would help in understanding the strengths and limitations of various approaches and could provide a more comprehensive justification for the chosen method.
2) Effectiveness of CCA-loss Across Different Models:
a) Question: The paper demonstrates the effectiveness of CCA-loss within the proposed model. Have you tested the applicability of CCA-loss with other LIC architectures or models? How does it perform in different settings?
b) Suggestion: Extending the evaluation of CCA-loss to other architectures could demonstrate its versatility and robustness. Including results from such experiments would show whether the benefits of CCA-loss are generalizable across different models.

**Limitations:**

1) The authors mention that current methods for organizing causal contexts are somewhat intuitive and that further exploration is needed. However, they do not delve deeply into the potential limitations this may impose on their results.
2) The implementation complexity of the proposed method, particularly the uneven channel-wise grouped strategy and CCA-loss function, is not explicitly discussed as a limitation.

---

> ### Author Rebuttal · Authors · 2024-08-07
>
> Thank you for your positive comments and insightful suggestions, which have significantly inspired us and enhanced our work. Our detailed feedback towards your comments is listed as follows.
>
> **Q1: Further Exploration of Causal Context Models (future research direction)**
>
> Leveraging decoded information to predict remaining information, i.e., conditional distribution modeling, plays a vital role in recent learned image compression (LIC) works. The previous works investigate different causal context architectures to make use of the naturally (indirectly optimized) dependency between learned representations, while, in this paper, we made the first attempt at imposing loss to adjust the causal context, which explicitly optimizes the predictability of the later information given the decoded information.
>
> In this paper, we validate our idea on the most commonly used causal context model, i.e., channel-wise autoregressive model. We hope our work could inspire follow-up works which not only investigate appropriate architectures for better modeling causal context but also pay attention to advanced learning strategy for obtaining better causal context model. To be more specific, with an advanced causal context adjusting capability, we think the encoder and entropy model in LIC should be designed together. For example, entropy models that use spatial context models should work together with the spatial-variant encoder; appropriate token coding order can be optimized in the token predicting framework, etc. We will follow your suggestion and conduct a more detailed analysis or ablation study on different causal context models.
>
> **Q2: Evaluate CCA-loss on more network architectures**
>
> We follow the reviewer's suggestion and replace the NAF-block in our paper with the residual block and the Swin-Transformer block, respectively. The compression results by different network architectures are shown in the following table, which clearly demonstrates the effectiveness of our proposed CCA-loss in improving the compression performance by different network architectures.
>
> |         Model          |   CCA Loss   | Inf. Time(ms) |            BD-rate            |
> | :-------------------- | :----------: | :-----------: | :--------------------------- |
> |  residual-block based  |              |      113      |            -15.02%            |
> |  residual-block based  | $\checkmark$ |      113      | -16.69% ( $\downarrow$ 1.67%) |
> | swin-Transformer based |              |      169      |            -16.46%            |
> | swin-Transformer based | $\checkmark$ |      169      | -17.67% ( $\downarrow$ 1.21%) |
> |    NAF-block based     |              |      116      |            -14.56%            |
> |    NAF-block based     | $\checkmark$ |      116      | -17.17% ( $\downarrow$ 2.61%) |
> |          BPG           |      -       |       -       |              0%               |
>
> **Q3: Complexity in Implementation**
>
> Thanks for your suggestion. We have tried our best to introduce our model clearly, and the uneven grouping strategy and the CCA-loss can be implemented with PyTorch by changing channel numbers in the entropy model and introducing auxiliary networks. Furthermore, we will release our code for implementation with easy-to-understand annotation after we finish the code cleanup.

---

> ### Comment · Reviewer_YBcc · 2024-08-12
>
> The author answered my doubts. I think the method proposed in this paper achieves excellent compression performance and contributes to the field of compression. I will keep my score.

---

### Official Review · Reviewer_cfcU · 2024-07-12

**Soundness:** 3
**Presentation:** 3
**Contribution:** 2
**Rating:** 5
**Confidence:** 3

**Summary:**

This paper proposed Causal Context Adjustment loss (CCA-loss) to explicitly adjust the causal context. However, this paper also proposes an efficient image compression model, which seems to be irrelevant to its main contribution, namely CCA loss. CCA loss and efficient architectures are orthogonal, and there is no obvious correlation between the two, which makes this paper rather scattered.

**Strengths:**

1. This paper proposed CCA loss to force the entropy model to learn better causal context organization.
2. This paper achieved the SoTA performance in the image compression task.
3. This paper is more efficient than previous methods, as reflected in its lower latency.

**Weaknesses:**

1. CCA Loss should assist more network architectures to prove its effectiveness. Using NAFNet alone is not enough.
2. This paper lacks analysis. Is the causal context organization of the model better after using CCA Loss? Some visual analysis can be given, for example, after using CCA Loss, can the model notice areas that the model without CCA Loss did not notice?

**Questions:**

Why use MS-SSIM instead of SSIM?

**Limitations:**

Please see the weaknesses and questions.

---

> ### Author Rebuttal · Authors · 2024-08-07
>
> Thank you for your comments, our detailed feedback towards your questions is listed as follows.
>
> **Q1: Evaluate CCA-loss on more network architectures**
>
> We follow the reviewer's suggestion and replace the NAF-block in our paper with the residual block and the Swin-Transformer block, respectively. The compression results by different network architectures are shown in the following table, which clearly demonstrates the effectiveness of our proposed CCA-loss in improving the compression performance by different network architectures.
>
> |         Model          |   CCA Loss   | Inf. Time(ms) |            BD-rate            |
> | :-------------------- | :----------: | :-----------: | :--------------------------- |
> |  residual-block based  |              |      113      |            -15.02%            |
> |  residual-block based  | $\checkmark$ |      113      | -16.69% ( $\downarrow$ 1.67%) |
> | swin-Transformer based |              |      169      |            -16.46%            |
> | swin-Transformer based | $\checkmark$ |      169      | -17.67% ( $\downarrow$ 1.21%) |
> |    NAF-block based     |              |      116      |            -14.56%            |
> |    NAF-block based     | $\checkmark$ |      116      | -17.17% ( $\downarrow$ 2.61%) |
> |          BPG           |      -       |       -       |              0%               |
>
> **Q2: The analysis on casual context distribution**
>
> The motivation of our proposed CCA-Loss is to adjust the causal context, making the latter representation more accurately predicted by the previously decoded representations. In our ablation study section, we have provided the information distribution ratios of different slices with and without our proposed CCA-loss (Figure 2 in our paper), and analyzed that our CCA-loss is able to push the network to encode significant information at earlier stages of the autoregressive model (lines 301-309). In order to address your concern, we further show **the coding bit map of the first three slices of *kodim06*, *kodim08* and *kodim13*, in our rebuttal PDF file**. The visualization examples clearly show the advantage of our CCA-loss. In our channel-wise autoregressive entropy model, CCA-loss is able to push the network to encode significant information at earlier slices and use them to better recover the latter slices. Overall, our model trained with CCA-loss could obtain a better rate-distortion trade-off.
>
> **Q3: MS-SSIM instead of SSIM as the eyesight evaluation**
>
> MS-SSIM is an extended version of SSIM, which calculates SSIM on multi-scales. In the literature of learned image compression, most recent works [4, 11, 14, 17, 20, 23, 25, 30, 37, 45, 49] adopt MS-SSIM as a loss or evaluation metric to train or evaluate the compression network. Therefore, in our paper, we follow the commonly adopted setting and use MS-SSIM instead of SSIM.

---

> > ### Comment · Reviewer_cfcU · 2024-08-08
> >
> > Thanks for the responses from the authors. This response solved the issues I raised, but I suggest that the authors add these supplementary experiments to the main text or appendix to enhance the paper's solidity. Overall, I intend to up my rating to 5.
> >
> > Also, authors could use LAM[1] to obtain clearer visualization results, further enhancing the persuasiveness of this paper.
> >
> > [1] Interpreting Super-Resolution Networks with Local Attribution Maps, CVPR 2021.

---

> > > ### Author Response · Authors · 2024-08-09
> > >
> > > Thank you for your response and the insightful suggestions in enhancing our paper. We will add these supplementary experiments to the appendix in our revised manuscript.

---

### Official Review · Reviewer_r1X1 · 2024-07-13

**Soundness:** 3
**Presentation:** 3
**Contribution:** 3
**Rating:** 5
**Confidence:** 5

**Summary:**

This work proposes a novel causal context adjustment loss to explicitly guide the encoder in prioritizing important information at the early stage of the autoregressive entropy model, which is both interesting and significant compared to the implicit modeling in ELIC. The loss is designed based on the entropy loss between different conditional channel-wise transformed latent slices. Extensive experiments have demonstrated the effectiveness of this design.

**Strengths:**

1. This work proposes a novel causal context adjustment loss to explicitly guide the encoder in prioritizing important information at the early stage of the autoregressive entropy model, which is both interesting and significant compared to the implicit modeling in ELIC.
2. The loss is designed based on the entropy loss between different conditional channel-wise transformed latent slices. Extensive experiments have demonstrated the effectiveness of this design.

**Weaknesses:**

There are some weaknesses that need to be addressed:

1. In Table 1, why is the performance only compared with anchor BPG instead of VVC? Whether this strategy is effective with stronger foundational compression codecs? It is suggested to validate your method with stronger codecs and compare the performance with VVC as shown in Table 2.
2. The abstract could be reorganized for a better illustration of the contribution. For example, you should highlight the explicit guidance for the encoder to adjust important information in the early stage of the autoregressive entropy model.
3. It would be better to compare with ELIC in lines 58-60.

**Questions:**

Whether this loss can be combined with the spatial context modeling strategy like "checkerboard"?

**Limitations:**

No limitation analysis is provided in this work.

---

> ### Author Rebuttal · Authors · 2024-08-07
>
> Thank you for your positive comments and insightful suggestions. We have conducted additional experiments, and our detailed feedback to your comments is listed below.
>
> **Q1: Ablation study on stronger codecs**
>
> In our submitted paper, we conducted ablation study with small model to facilitate our experiments. To address your concern about whether our proposed CCA-loss is helpful for stronger compression models, we further conducted an ablation study on our large model. We retrained our model without the proposed CCA-loss and reported the BD-rate in the following table. Moreover, we follow your request and use VVC as the baseline in the table. As shown in the table, our proposed CCA-loss is beneficial for both small and large models.
>
> |    Model     |   CCA Loss   | Inf. Time(ms) |            BD-rate             |
> | :---------- | :----------: | :-----------: | :---------------------------- |
> | Ours (large) |              |      201      |            -12.39%             |
> | Ours (large) | $\checkmark$ |      201      | -13.87% ( $\downarrow$ 1.48% ) |
> | Ours (small) |              |      116      |             4.78%              |
> | Ours (small) | $\checkmark$ |      116      |  1.24% ( $\downarrow$ 3.54% )  |
> |     VVC      |      -       |       -       |               0%               |
>
> **Q2: Revision suggestions (Abstract, Compare with ELIC in lines 58-60)**
>
> We concur with the reviewer’s observation that there exists some inadequacy in our abstract and introduction sections. we would reorganize the abstract section and add the comparison with ELIC to the introduction section in our revised manuscript.
>
> **Q3: Combining with spatial context modeling**
>
> As we have stated in our paper (lines 234 - 236), although our proposed CCA-loss is able to guide the encoder to adjust the causal context, the convolutional encoder used in our work can only extract information in a spatially invariant manner. Therefore, for commonly used spatial context models, such as the checkerboard model, our current model is unable to adjust the causal context to improve the compression performance. In order to boost the spatial context model with our proposed CCA-loss, we need to have a tailored encoder which could adjust representation according to spatial position. In the future, we will explore whether we could propose a new spatial context model for Transformer-based encoder, which uses decoded tokens to estimate the remaining tokens; in that case, we believe that our proposed CCA-loss could play a positive role in learning better spatial context model.

---

> > ### Comment · Reviewer_r1X1 · 2024-08-08
> > **Response to authors**
> >
> > Thanks for your responses from the authors. Can you clarify the differences between your large model and your small model? Generally, a simple model size cannot bring such performance improvement in compression actually. What's the improvement are from? Transform? Quantization? Entropy Coding?

---

> > > ### Author Response · Authors · 2024-08-08
> > >
> > > We sincerely appreciate your response. The differences between the large and small models lie in the encoder-decoder transform as well as entropy model. The large model has 64.89M parameters and 615.93G FLOPs, while the small model has 22.16M parameters and 150.55G FLOPs. Specifically, compared to the large model, we have removed the residual blocks in both the encoder and decoder, and reduced the number of channels in the encoder and decoder from [192, 224, 256] to [128, 128, 128]. Additionally, the NAF-block stacks in the auto-encoder and entropy models contain 4 blocks for the large model and 2 blocks for the small model. The autoregressive slices are set to 5 for the large model and 3 for the small model. The stronger encoder-decoder in the large model produces a more robust latent representation, and the larger entropy model more accurately estimates the probability distribution, leading to better rate-distortion performance, which accounts for the performance gap between the two models. We will release the code and checkpoints for both models after completing the code cleanup.
> > >
> > > | Model | #Params | FLOPs(G) | Channel number  | NAF-blocks | AR slices | Residual blocks |
> > > | :---: | :-----: | :------: | :-------------: | :-------: | :-------: | :-------------: |
> > > | large | 64.89M  |  615.93  | [192, 224, 256] |     4     |     5     |  $\checkmark$   |
> > > | small | 22.16M  |  150.55  | [128, 128, 128] |     2     |     3     |        X        |

---

> > > > ### Comment · Reviewer_r1X1 · 2024-08-12
> > > > **Response to authors**
> > > >
> > > > Thanks for the responses of authors. I will keep my score.

---

### Official Review · Reviewer_NpdM · 2024-07-14

**Soundness:** 3
**Presentation:** 3
**Contribution:** 3
**Rating:** 7
**Confidence:** 4

**Summary:**

The paper proposes an auxiliary neural network in conjunction with a loss  function during training to improve the better contextual modeling of the data during training phase. The advantage of the proposed technique is that the auxiliary network is not required during inference and hence the complexity is minimal. The work conducts a series of experiments that supports the claims.

**Strengths:**

* The paper is well written.
* The approach to bring in context information into play is relatively novel.
* Experiments are convincing and covers the necessary and sufficient datasets.
* The ablation study supports the claims.
* Gain versus FLOP calculation is good.

**Weaknesses:**

* There is no strong weakness

**Questions:**

* Could you elaborate on MAC operations versus FLOPS?

**Limitations:**

No discussion on limitations

---

> ### Author Rebuttal · Authors · 2024-08-07
>
> Thank you for your positive comments.
>
> Multiply–accumulate operations (MACs) and floating point operations (FLOPs) are two important measurements of model computation. More specifically, MACs calculate the number of multiply-accumulate operations, and each MAC operation consists of a multiplication followed by an addition. While, FLOPs calculate the number of floating-point operations, including all types of floating-point computations such as addition, subtraction, multiplication, and division. In the literature of learned image compression (LIC), the number of FLOPs is reported as the metric of the complexity of the model. In our submitted paper, we followed the commonly adopted setting and reported the FLOPs number to compare with other methods. In order to answer your question, we recalculated the number of MACs and FLOPs by different methods and reported in the following table. As can be found in the table, our proposed method could achieve state-of-the-art compression results with less computational footprint.
>
> |        Model        | Enc. Time | Dec. Time | #Params | BD-rate | FLOPs(G) | MACs(G) |
> | :----------------- | :-------: | :-------: | :-----: | :-----: | :------: | :-----: |
> | Zou et al. (CVPR2022) |   248ms   |   176ms   | 99.83M  | -4.01%  |  200.11  | 199.94  |
> | Zhu et al. (ICLR2022) |   129ms   |   143ms   | 56.93M  | -3.00%  |  364.08  | 209.40  |
> | Liu et al. (CVPR2023) |   122ms   |   133ms   | 75.90M  | -11.88% |  700.65  | 702.81  |
> |        Ours         |   109ms   |   92ms    | 64.89M  | -13.87% |  615.93  | 492.65  |
> |         VVC         |     -     |     -     |    -    |   0%    |    -     |    -    |

---

### Author Rebuttal · Authors · 2024-08-07

We are sincerely grateful to all the reviewers for their valuable time and expert insights; we truly appreciate their diligent work and constructive feedback.

---

### Decision · Program_Chairs · 2024-09-25

**Decision:**

Accept (poster)

**Comment:**

While there were some reservations in the initial reviews, the rebuttal addressed most of the reviewers' concerns. All the reviewers unanimously voted for acceptance in their final rating. The AC agrees with the reviewers' recommendations and supports acceptance of the paper. The AC recommends including additional evaluations performed during the discussion of the paper (either in the appendix or main paper, space permitting.).